# Effects of Ultra-Sonication and Agitation on Bioactive Compounds and Structure of Amaranth Extract

**DOI:** 10.3390/foods9081116

**Published:** 2020-08-13

**Authors:** Maruf Ahmed, Karna Ramachandraiah, Gui-Hun Jiang, Jong Bang Eun

**Affiliations:** 1Department of Food Processing and Preservation, Hajee Mohammad Danesh Science and Technology University, Dinajpur 5400, Bangladesh; maruf@hstu.ac.bd; 2Department of Food Science and Biotechnology, Sejong University, Seoul 05006, Korea; karna@sejong.ac.kr; 3School of Public Health, Jilin Medical University, Jilin, Changchun 130026, China; jiangguihun1@naver.com; 4Department of Food Science and Technology and BK 21 PlusProgram, Graduate School of Chonnam National University, Gwanju 61186, Korea

**Keywords:** Amaranth, ultra-sonication, agitation, Fourier-transformed infrared, Antioxidants properties

## Abstract

Amaranth is an excellent source of various bioactive compounds that could be beneficial in the prevention of some human diseases. This study investigated the extraction and characterization of bioactive compounds from amaranth using ultra-sonication and agitation at 30, 50 and 70 °C. Color L* values showed significant (*p* < 0.05) differences at 70 °C between ultra-sonication and agitation. Ultra-sonication temperature had significant effect on L* and a* values whereas agitation temperature did not have a significant effect on L*, a* and b* values. No significant (*p* < 0.05) differences were found in terms of total phenol, total flavonoid, DPPH^•+^, ABTS^+^ scavenging activity, betacyanins, betaxanthin and betanicaicd between ultra-sonication and agitation. However, temperature had a significant (*p* < 0.05) effect on total phenol (8.64–10.598 mg/g), DPPH^+^scavenging activity (84.36–94.44%), betacyanins (4585.95–5325.32 mg/100 g), betaxanthin (1312.56–1524.06 mg/100 g) and betalamic acid (1408.15–1790.22 mg/100 g) in ultra-sonication. Higher temperature (70 °C) showed greater amount of arbutin and hydroxybenzoic acid than those of lower temperature (30 °C) for both extraction methods. Meanwhile, temperature did not affect vanillic acid, *p*-coumaric acid and ferulic acid for both samples. Fourier-transformed infrared (FTIR) spectrometry showed that ultra-sonication and agitation resulted in similar effect on the structure of amaranth extracts. Higher temperature was correlated with bioactive compounds, which were observed by principal component analysis (PCA). Therefore, agitation at 70 °C could be used as an alternative for ultra-sonication to improve the bioactive compounds and antioxidant activities of amaranth. In addition, agitation and ultra-sonication techniques might be served as an alternative of conventional technique.

## 1. Introduction

Amaranthus (family *Amaranthaceae*) is a major source of vitamins, such as folic acid, protein, dietary fiber, and minerals [1]. This underutilized plant is also known to contain valuable bioactive compounds that include betacyanins, betaxanthins, and polyphenols, which can inhibit deteriorative diseases that include cardiovascular disorders and inflammatory responses [2]. In particular, amaranthus leaves contain several types of betalains such as amaranthin, isoamranthin, methyl derivative of arginine betaxanthin and betamic acid [2]. Furthermore, different types of phenolic compounds such as gallic acid, chlorogenic acid, ferulic acid, salicylic acid, rutin and quercetin have also been studied in many Amaranthus species [3]. A few Amaranthus species have also shown strong antioxidant and anti-proliferative activity on Ehrlich’s ascites carcinoma cells [4]. Amaranth bioactive compounds have been used as abundant sources of natural bioactive compounds and natural food colorant [5]. Natural pigments have been widely used as natural preservatives in cosmetic products, drugs and food [6]. Various potential applications such as composite cookies, gluten-free bread and juice can be produced using amaranth leaves and seeds [7].

Antioxidants derived from natural sources such as fruits and vegetables have been gaining popularity due to various health benefits. However, the perishable nature and lower shelf-life necessitate fruits and vegetables to be processed. The traditional methods of processing involve extraction of bioactive compounds using solvents such as water, methanol and ethanol [8,9,10]. Recently, ultrasound-assisted and agitation extractions are considered to be more efficient for the derivation of valuable bioactive compounds from different sources. Particularly, ultrasound has been shown to promote the release of flavonoid compounds in raspberry and blueberry puree due to the disruption of the cell wall [11]. Since the amount of cavitation bubbles is contingent upon temperature, it is likely that increased ultrasonic temperature may improve cavitation and thereby cause cell rupture. Nonetheless, ultrasound frequency and power could have a negative impact on flavonoid content due to oxidation [12]. It has been reported that, apart from ultrasonic extraction temperature, time also influences the extraction of flavonoid from pumpkin [13]. Longer ultrasonic extraction time was reported to decrease the flavonoid content in vegetables [14]. Ultrasound extraction techniques used as rapid and green extraction process have been utilized for the derivation of bioactive compounds from natural sources [15]. Agitation was also shown to improve the extraction of phenolic compounds from olive leaves due to the rupture of the cell and the subsequent release of phenolic compounds from the cell matrix [16].

Color is an important parameter that influences the acceptance of foods and beverages. In the food industry, synthetic colorants are widely used to impart desired colors to food products. However, studies have shown that many synthetic food colorants likely to be carcinogenic to consumers [17]. Therefore, processors are utilizing natural colorants as a viable alternative. In this regard, betalains from red amaranth could be an alternative source of natural colorant and antioxidant compounds. Owing to such benefits, this polyphenol has received great attention by food manufacturers in recent times [8]. However, the investigations on the extraction of antioxidant from red amaranthus leaves using ultra-sonication and agitation are limited. Therefore, the objective of this study was to investigate the effects of ultra-sonication and agitation on phenolic compounds from red amaranth. In addition, chemical bonds in the extract samples were identified by using Fourier-transform infrared (FTIR). Furthermore, principal component analysis (PCA) was used to establish the correlation between nutritional components and extraction techniques.

## 2. Materials and Methods

### 2.1. Sample Collection and Preparation

Fresh red amaranth (*Amaranthuscruentus*) was procured from the local market, Dinajpur, Bangladesh. The plant materials were washed in running tap water to eliminate any dirt, and other surface impurities. Using a stainless-steel knife, the collected samples (approximately 500 to 800 g) were then cut into small pieces and the leafy parts were separated from the roots. The freshly cut small pieces (leafy parts) were placed in a dryer at 40 °C for 25 h. The dried red amaranth samples were pulverized using a blender and stored at 4 °C until further analysis.

### 2.2. Chemical and Reagents

2,2-azino-bis (3-ethylbenzthiazoline-6-sulfonic acid) (ABTS^+^), 1,1-diphenyl-2-picrylhydrazyl (DPPH^+^), rbutin, Ferulic acid, *p*-coumaric acid, Hydroxybenzoic acid, Vanillic acid, and Folin-Ciocalteu reagent, were obtained from Sigma-Aldrich, Chemical Co. (St. Louis, MO, USA). Other chemicals, namely, sodium carbonate, sodium chloride, aluminum chloride, sodium nitrite, and lead acetate, used in the present study were of analytical grade.

### 2.3. Extraction of Amaranth Powder

#### Ultra-Sonication and Agitation Extraction

In particular, 6.25 g of amaranth powder was taken in a 250-mL conical flask and mixed with 100 mL of distilled water and placed in an ultra-sonicator bath (Bandelin Sonorex, RK510H, 35 kHz, Germany) and agitated using a shaking water bath (JSSB-50T, South Korea) at 100 rpm, respectively, at 30, 50 and 70 °C for 5 min. The extracted sample was filtered through a cheese cloth, and then vacuum filtered through a Whatman filter paper No.1. The quality parameters of the filtered extract were then analyzed.

### 2.4. Hunter Color Values

Das et al. [18] method was followed to determine the color of the amaranth extracts using a Chroma meter (Minolta, CR-300, Osaka, Japan). Readings were expressed as L*, a* and b* parameters.

### 2.5. Determination of Amaranthus Pigments

Amaranthus pigments were quantified using the modified method described by Kumar et al. [9]. In particular, a 1 mL aliquot was diluted with 9.0 mL of distilled water and absorbance was measured at 538, 480 and 430 nm for betacyanins, betaxanthins and betalamic acid, respectively, using a spectrophotometer (Optizen 2120 UV, Mecasys Co., South Korea). The pigment content was quantified based on the following equation.
(mg/100 g of dry matter) = (A × MW × V × DF × 100)/(ε × L × W)(1)
where A = Absorbance; MW = Molecular weight of betacyanins; (726.6) betaxanthins (309) and betalamic acid (212); V = Solution volume; DF = Dilution factor; ε = Molar extinction coefficient of betacyanins (5.66 × 10^4^ M^−1^ cm^−1^), betaxanthins (48,000 M^−1^ cm^−1^), and betalamic acid (24,000 M^−1^ cm^−1^), W = Sample weight (g).

### 2.6. Determination of Total Phenol Content

The total phenol content of the amaranth sample was measured according to the method described by Lee et al. [19] with some modification. Amaranth extract (1 mL) was diluted 10 times with distilled water and 200 μL samples along with 1.5 mL of 10% Folin-Ciocalteu reagent in test tubes for 5 min. Following the 1.5-mL addition of sodium carbonate (7.5%), the samples were incubated for 30 min in the dark at room temperature. A 725-nm wavelength was used to measure the absorbance the solution using a UV/Vis spectrophotometer (Optizen 2120 UV, South Korea); Gallic acid was also prepared for the calibration and results were expressed as mg per gram sample (mg/g).

### 2.7. Determination of the Flavonoid Content

Hajimahmoodi et al. [20] method was followed for the determination of flavonoid content Briefly, 1 mL diluted sample (5 times) was treated with 4 mL distilled water along with 0.3 mL of 5% sodium nitrite in a test tube for 5 min. Following incubation, each tube was treated with 0.3 mL of aluminum chloride and incubated for 6 min at room temperature. Then, 2 mL sodium hydroxide was mixed and absorbance was recorded at a 510-nm wavelength using a UV/Vis spectrophotometer. Rutin was used for the preparation of standard curve and results were expressed as mg per gram sample (mg/g).

### 2.8. Determination of Phenolic Compounds Using High-Performanceliquid Chromatography (HPLC)

Phenolic compounds were determined according to the method described by Yim and Nam [21] using a Chromatography instrument (LC-20 Avp Shimadzu Co., Japan) along with a diode array detector. Separation of phenolic compounds was done by using A C18 HPLC column (300 × 3.9 mm). The mobile phases were constituted by 2% (*v/v*) aqueous acetic acid (solvent A) and 0.5% (*v/v*) acetic acid in 50% acetonitrile (solvent B) with the following linear gradient: 2% B from 0 to 5 min; 2% to 55% B from 5 to 55 min; 100% B from 55 to 65 min;10% B from 65 to 70 min; and post run with 2% B for 6 min. Then, 0.45-μm filters were used to filter the samples and 20 μL of samples, and standard solutions were injected into the HPLC system. The flow rate was 1 mL/min at 30 °C. Chromatograms were recorded at 280 nm for arbutin, hydroxybenzoic acid and vanillic acid and at 320 nm for *p*-coumaric acid and ferulic acid. Five concentrations (20 to 100 ppm) were used to prepare the calibration curve and regression equations were found by plotting the area of the standard solutions against concentrations. The quantities of phenolic compounds were calculated by comparing the retention time of the samples with those of the standard solutions. Data were shown as mg per gram sample (mg/g).

### 2.9. Antioxidant Capacity Using DPPH^+^ Assay

DPPH^+^ assay was performed according to the method of Dong et al. [22] with some modifications. In particular, a stock solution was prepared with 24 mg of DPPH^+^ using methanol. The stock solution (10 mL) was then diluted with methanol to form the working solution. Water was used to dilute (5 times) the Amaranth extract (1 mL) and samples (200 μL) were mixed with the DPPH^+^ solution (2 mL). Following a 30-min incubation, the absorbance of the solution was recorded at 515 nm using a UV/Vis spectrophotometer. The following formula was used to calculate the DPPH^+^ radical-scavenging activity:Scavenging activity (%) = (1 − A_1_/A_0_) × 100(2)
where A_0_ = absorbance of control solution and A_1_ = absorbance of sample solution. Sample absorbance was calculated without the absorbance of DPPH^+^ solution.

### 2.10. Antioxidant Capacity Using ABTS^+^ Assay

The ABTS^+^ assay was performed as described by Dong et al. [22] with minor modifications. Briefly, 7.4 mM ABTS^+^ solution potassium persulfate (2.6 mM) was prepared in equal amounts and placed in the dark (12 h). Methanol was used to dilute the solution until an absorbance of 1.1 ± 0.04 at 734 nm was attained. Amaranth samples (1 mL, diluted 5 times in water) were incubated with 2 mL ABTS^+^ solution for 10 min and the absorbance was recorded at 734 nm. ABTS^+^ scavenging value was calculated based on the following formula:Scavenging activity (%) = (1 − A_1_/A_0_) × 100(3)
where A_0_ = absorbance of control solution (without sample) and A_1_ = absorbance of sample solution. Sample absorbance was quantified by subtracting the sample absorbance values that was incubated without ABTS^+^ solution.

### 2.11. Fourier-Transformed Infrared (FTIR) Spectrometry

Chemical bonds of amaranth extracts were measured by Perkin Elmer FTIR spectrophotometer (Perkin Elmer, Inc., Waltham, MA, USA). Sample was kept on universal diamond ATR top-plate and then 120 N was applied on the top of the sample. A 380- to 4000-cm^−1^ wavelength at a 4-cm^−1^ resolution with four scans was also used. Alcohol was used to clean the universal diamond ATR top-plate.

### 2.12. Statistical Analysis

Phenolic compounds were measured twice whereas other parameters were carried out thrice. Statistical software (SPSS for Windows Version 21.0) was used to carry out the one-way analysis of variance (ANOVA) using the Duncan’s test at 5% significance level. Results expressed as mean vale ± standard deviation (SD). Principal Component Analysis (PCA) was performed using XLSTAT 2017.

## 3. Results and Discussion

### 3.1. Effects of Ultra-Sonication and Agitation on Color Values of Amaranthus Extracts

L*, a* and b* values of amaranth extracts obtained by means of ultra-sonication and agitation temperatures were shown in Figure 1. Higher L* values were observed with ultra-sonication samples than with that of agitation samples. Significant (*p* < 0.05) differences in L* values were observed between ultra-sonication and agitation samples at higher temperatures. Ultra-sonication temperature had significant effect on L* and a* values whereas agitation temperature did not exhibit any significant effect on L*, a* and b* values. On the other hand, b* values did not show any significant differences between ultra-sonication and agitation samples. The changes in color might be related to betacyanin content and various isomerized forms such as betaxanthin and betalamic acid. A good correlation was obtained between the L* values and the amaranth pigments (*r* values of 0.80, 0.80, and 0.88 for betacyanins, betaxanthin and betanic acid, respectively, at *p* < 0.05). However, no correlation was observed between a* and b*values and the amaranth pigments. While L* and a* values were much lower, b* values were found to be much higher than that of powders derived from different parts of the amaranthus species [3]. The deviation can be attributed to differences in cultivar and processing environments [3].

### 3.2. Effects of Ultra-Sonication and Agitation on Anti-Oxidative Properties

The anti-oxidative properties of amaranthus extracts prepared using ultra-sonication and agitation were shown in Figure 2 and Figure 3. Higher temperature (70 °C) resulted in increased (*p* < 0.05) total phenol and total flavonoid contents in ultra-sonication than with lower temperature (30 °C). On the other hand, agitation temperature did not significantly affect total phenol and total flavonoid contents. The highest total phenol (10.598 mg/g for ultra-sonication and 95.40 mg/g for agitation) and total flavonoid contents (5.559 mg/g for ultra-sonication and 5.324 mg/g for agitation) were obtained at 70 °C. The total phenol and total flavonoid content quantified in this study were comparable with the different parts of amaranthus species, as reported by Li et al. [3]. The higher phenol and flavonoid contents were observed at a higher temperature (70 °C) due to the release of bound polyphenols. It has been reported that bound polyphenols are released upon disruption of the cell wall [23]. It is likely that the cell wall could have been disrupted by higher temperature. The increased polyphenolic oxidase activity followed by sonication could have also affected the total phenol and total flavonoid content [24]. Total phenol and total flavonoid content could be enhanced by sonication in various juices [25,26] and plants [27] have been reported. Ultra-sonication samples showed higher DPPH^+^ and ABTS^+^ scavenging activity than those of agitation samples (3A and 3B). For ultra-sonicated samples, the two radical scavenging activities increased with higher extraction temperature. Significant (*p* < 0.05) differences were found between lower and higher extraction temperature for ultra-sonication sample. On the other hand, different agitation temperatures did not influence the radical scavenging activities. The surge in DPPH^+^ and ABTS^+^ scavenging activity could be due to elevated levels of total phenol and total flavonoid contents. Higher correlation was observed between total phenol and ABTS^+^ (*r* value of 0.98 at *p* < 0.05) than between total phenol and DPPH^+^ (*r* value of 0.78 at *p* < 0.05). Similar correlations were found for total phenol and total flavonoid contents with the antioxidant capacity [25,26]. The same trend was not found for DPPH^+^ and ABTS^+^ for the techniques of ultra-sonication and agitation. That DPPH^+^ showed higher values than ABTS^+^ could be due to DPPH^+^ acting as a stronger electron donor than ABTS^+^ [28].

### 3.3. Effects of Ultra-Sonication and Agitation on Betacyanins, Betaxanthins and Betalamic Acid

The amount of extracted betacyanins, betaxanthin and betalamic acid ranged from 4585.95 to 5325.32 mg/100 g, 1312.56 to 1524.06 mg/100 g and 1408.15 to 1790.22 mg/100 g, respectively, for ultra-sonication samples. Agitation extracted samples ranged from 4536.59 to 4758.42 mg/100 g for betacyanins, 1271.36 to 1317.37 mg/100 g for betaxanthin, and 1310.63 to 1342.2 mg/100 g for betalamic acid (Figure 4). Higher extraction of betacyanins, betaxanthin and betalamic acid could be due to greater of cell wall by ultra-sonication compared to agitation. However, these values were higher than those observed in other studies on edible portions of amaranthus seed (0.07 to 0.96 mg/100 g), amaranthus stalks (0.56 to 1.54 mg/100 g), amaranthus leaves (16.90 to 20.93 mg/100 g), amaranthus flowers (0.95 to 6.02 mg/100 g) and amaranthus sprouts (2.69 mg/100 g) [3]. These variations could be due to dissimilar methods of extraction and origin of samples [3]. In this study, higher temperature showed higher betacyanins, betaxanthin and betalamic acid compared to lower temperature for both extraction methods. Temperature significantly (*p* < 0.05) affected the betacyanins, betaxanthin and betalamic acid when ultra-sonication was employed. This might be due to increased extractability of betacyanins, betaxanthin and betalamic acid at higher temperatures. On the other hand, agitation temperature had no major impact on amaranth pigments. The three pigments followed similar trend to that of total phenolic content. In a study by Gokhale and Lele [29] on beet, betacyanin decreased but betaxanthin increased with increasing extraction temperatures. Apart from being a different sample, different extraction temperatures could have caused such differences. However, a strong correlation was found between betacyanin content and total phenol (*r* value of 0.97 at *p* < 0.05), betaxanthin content and total phenol (*r* value of 0.90 at *p* < 0.05) and between betalamic acid and total phenol (*r* value of 0.86 at *p* < 0.05). On the other hand, correlation was also found between betacyanin content and ABTS^+^ (*r* value of 0.94 at *p* < 0.05, betaxanthin content and ABTS^+^ (*r* value of 0.87 at *p* < 0.05) and between betalamic acid and ABTS^+^
*r* value of 0.83 at *p* < 0.05).

### 3.4. Effects of Ultra-Soniation and Agitation on Individual Phenolic Compounds

Individual phenolic compounds in amaranthus extracts as influenced by ultra-sonication and agitation were shown in Table 1. In this study abundant phenolic compounds found in ultra-sonicated samples were arbutin > hydroxybenzoic acid > ferulic acid > *p*-coumaric acid >vanillic acid whereas agitated samples contained arbutin > hydroxybenzoic acid > vanillic acid > ferulic acid > *p*-coumaric acid. The changes could be attributed to different extraction methods. In a study by Li et al. [3] individual compounds found were gallic acid, protocatechuic acid, cholorogenic acid, getistic acid, hydroxylbenzoic acid, ferulic acid, rutin and quercetin in different parts of amaranthus species. Paśko et al. [30] found several phenolic compounds such as gallic acid, hydroxybenzoic acid, vanillic acid, *p*-coumaric acid, syringic phenolic acids in seeds and sprouts of amaranth. Different extraction methods and origin of samples might be the reason to get the variations of this result from others results. In this study, higher amounts of phenolic compounds were obtained in agitation as compared to ultra-sonication. Higher temperature (70 °C) significantly increased arbutin and hydroxybenzoic acid as compared to lower temperature (30 °C) in ultra-sonication methods. On the other hand, in agitation methods, only arbutin increased significantly but not hydroxybenzoic acid at higher temperature (70 °C) than at lower temperature (30 °C). However, vanillic acid, *p*-coumaric acid and ferulic acid remained unaffected irrespective of temperature for both the methods. Agitation retained higher amount of total phenolic compounds than those of ultra-sonication techniques at higher temperature. Increases in arbutin and hydroxybenzoic acid might be extracted from bound polyphenols, as studies have shown that bound polyphenols might be released the disruption of the cell wall [29]. Sonication and agitation have ability to disrupt the cell wall [24,25]. The reduction in vanillic acid and ferulic acid could be due to enzymatic degradation or co-pigmentation reaction [30]. In a study by Mphahlele et al. [31] on pomegranate peel, a higher amount of rutin, *p*-coumaric, catechin, and epicatechin but lower amounts of hesperid were observed at 60 compared to 30 °C. In this study, the amounts of vanillic acid (35 to 158.4 µg/g), ferulic acid (73.8 to 93.76 µg/g), *p*-coumaric acid (28.80 to 69.91 µg/g) and hydroxybenzoic acid (1519.40 to 2183.88 µg/g) measure were consistent with that of Venskutonis and Kraujalis [32], who studied various amaranth species. However, the hydroxybenzoic acid level (1519.40 to 2183.88 µg/g) was much higher than amaranthus seeds and sprouts (8 to 20 µg/g) [23]. There is no information available about arbutin in amaranth. Nonetheless, cultivated species and different extraction methods might be reason to get the variation results from other researchers. 

### 3.5. Structural Changes

FTIR analysis was performed to ascertain any structural changes induced by ultra-sonication and agitation treatments on amaranth extract. FTIR spectra of the amaranth extracts were shown in Figure 5a–f. Carbonyl (C=O) and amines group (N-H) are considered to represent the betacyanin group of betalains pigments [33]. The intensity of bands at 1636.69 to 1637.49 cm^−1^; 2088.96 to 2125.76 cm^−1^; 3309.94 to 3311.54 cm^−1^ was observed for agitation extracted samples whereas the peak at 1637.33 to 1637.71 cm^−1^; 2097.60 to 2099.14 cm^−1^; 3292.59 to 3309.80 cm^−1^ was identified for the ultra-sonication extracted samples. Skenderidis et al. [34] found the peak at 1720 and 3315 cm^−1^ to be responsible for the carboxyl group and hydroxyl group, respectively, in the ultrasound-assisted extraction of lyophilized powdered pomegranate peel. Biswas et al. [2] also mentioned that the 1653 to 918 cm^−1^ band represented the functional group of betacyanin in amaranth tricolor pigments. Cai et al. [35] also showed the presence of functional group of betacyanin at 1720–714 cm^−1^ in *Amaranthus* pigments. These variations may be due to the C=O and N-H group. On the other hand, the absorption bands at 3264 to 3311.54 cm^−1^ were found in amaranth extract. These regions represent the -NH2, -OH, -H_2_O and C-S functional group. Functional group could vary due to protein, amino acid and water in the pigments [2]. In this study, FTIR results showed that ultra-sonication and agitation resulted in similar effect on structure of amaranth extracts.

### 3.6. Principal Component Analysis (PCA)

Relationship between data and samples were identified by PCA in this study. The results indicated that components PC1 (58.65%) and components PC2 (23.55%,) were accountable of the total variance and presented Eigen values 8.79 for PC1 and 3.53 for PC2, which were greater than 1.0. The results showed that all phenol, flavonoid, betacyanins, betaxanthins and betalamic acid were associated with ultra-sonication (US) 70 °C whereas DPPH^+^, ABTS^+^, arbutin and hydroxybenzoic acid were associated with water bath (WB) 70 °C. These phenomena might be related to the amount of compounds. Higher negative scores were correlated with PC1 along with phenol, flavonoid, DPPH^+^, ABTS^+^ betacyanins, betaxanthins betanic acids, arbutin, hydroxybenzoic, ferulic acid (Table 2, Figure 6). On the other hand, higher positive scores were associated with PC2 along with arbutin, hydorxybenzoic acid, vanillic acid and ferulic acid (Table 2 and Figure 5). However, negative scores were observed with L*and b* in PC2 (Table 2). Therefore, results showed the correlations between data and samples using principal components analysis that could be useful to choosing the extraction method.

## 4. Conclusions

The effects of ultra-sonication and agitation extractions at 30, 50 and 70 °C on bioactive compounds, antioxidant activities and structure changes of amaranth extract were investigated. No significant differences were observed in total phenol, total flavonoid, DPPH^+^, ABTS^+^ scavenging activity, betacyanins, betaxanthin and betalamic acid of amaranth extracts prepared using ultra-sonication and agitation. A higher temperature led to increased total phenolic compounds in the amaranth extracts. Both extraction methods did not affect the structures, which were analyzed using FTIR. A connection was established between the extraction methods and the nutritional components by principal component analysis. Therefore, agitation extraction at 70 °C could be considered as an efficient method, which could also serve as an alternative for ultra-sonication in the derivation of bioactive compounds from amaranth extracts.

## Figures and Tables

**Figure 1 foods-09-01116-f001:**
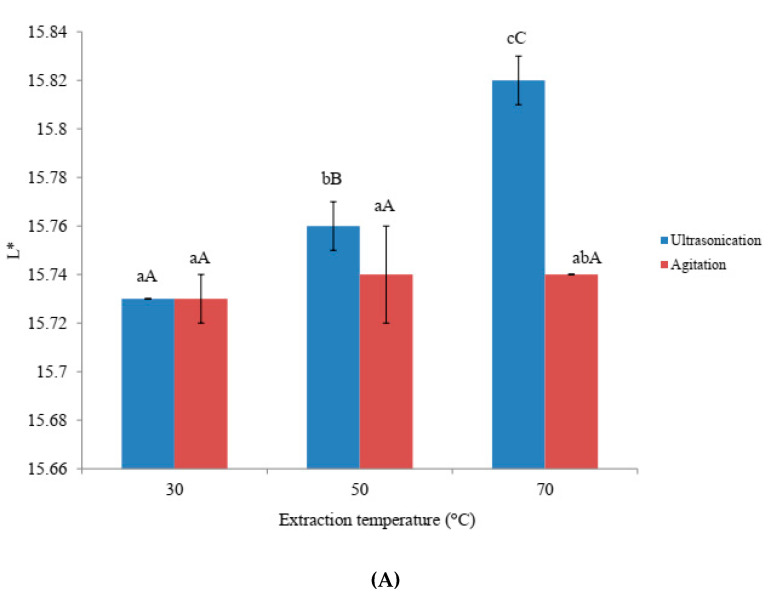
Effects of ultra-sonication and agitation on color values of amaranth extract. (**A**) L* values (**B**) a* values (**C**) b* values. ^a–d^ Means with same subscript alphabets are not significantly different (*p* ≤ 0.05) between each extraction method. ^A–C^ Means with same subscript alphabets are not significantly different (*p* ≤ 0.05) among different temperatures in each extraction method.

**Figure 2 foods-09-01116-f002:**
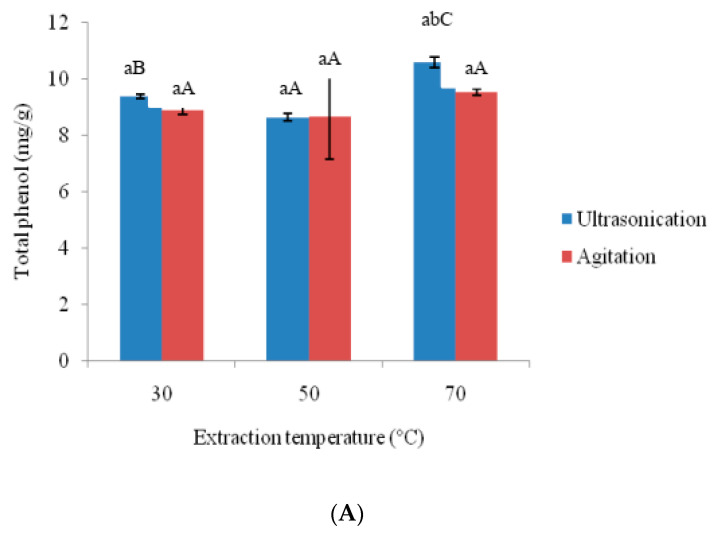
Effects of ultra-sonication and agitation on total phenol and total flavonoid content of amaranth extract. (**A**) Total phenol content (**B**) Total flavonoid content. ^a–d^ Means with same subscript alphabets are not significantly different (*p* ≤ 0.05) between each extraction method. ^A–C^ Means with same subscript alphabets are not significantly different (*p* ≤ 0.05) among different temperatures in each extraction method.

**Figure 3 foods-09-01116-f003:**
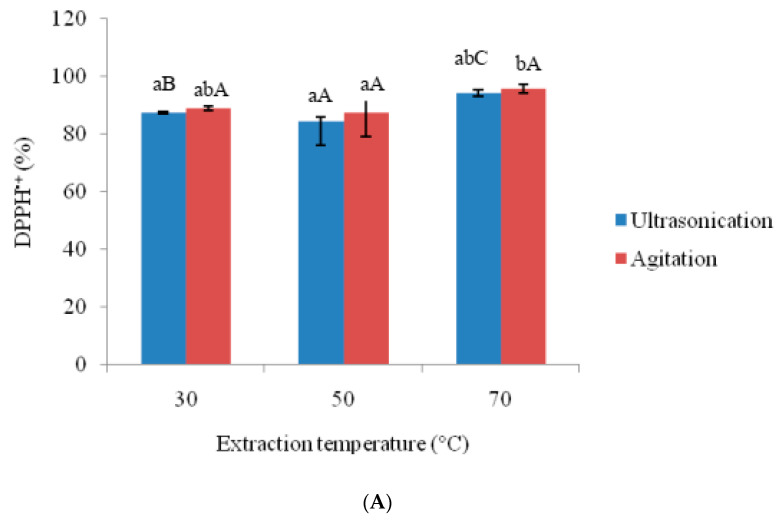
Effects of ultrasonication and agitation on antioxidant activities of amaranth extract. (**A**) DPPH^+^ (**B**) ABTS^+^. ^a–b^ Means with same subscript alphabets are not significantly different (*p* ≤ 0.05) between each extraction method. ^A–C^ Means with same subscript alphabets are not significantly different (*p* ≤ 0.05) among different temperatures in each extraction method.

**Figure 4 foods-09-01116-f004:**
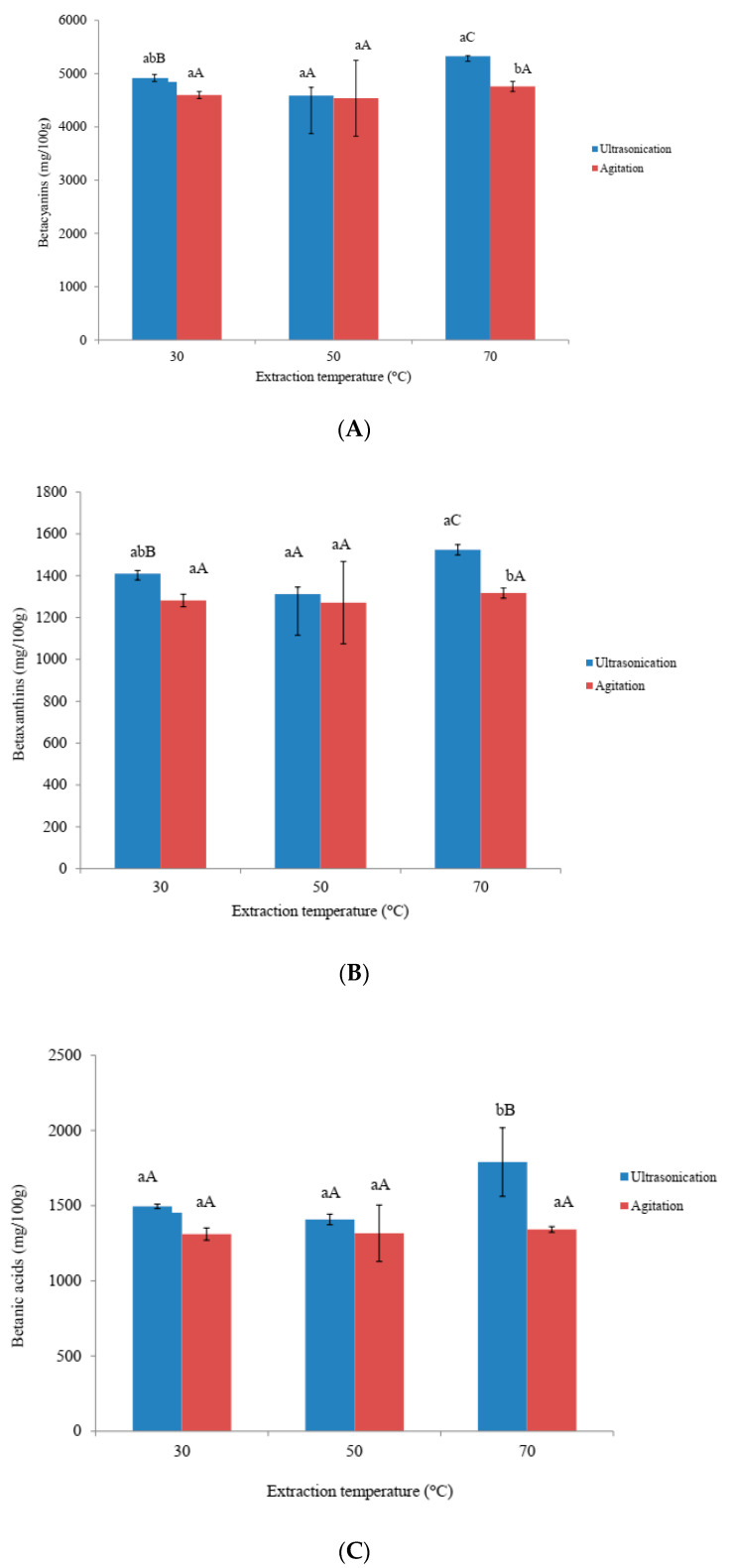
Effects of ultra-sonication and agitation on Betacyanins content, Betaxanthins content and Betanic acids of amaranth extract. (**A**) Betacyanins content (**B**) Betaxanthins content (**C**) Betalamic acid. ^a–b^ Means with same subscript alphabets are not significantly different (*p* ≤ 0.05) between each extraction method. ^A–C^ Means with same subscript alphabets are not significantly different (*p* ≤ 0.05) among different temperatures in each extraction method.

**Figure 5 foods-09-01116-f005:**
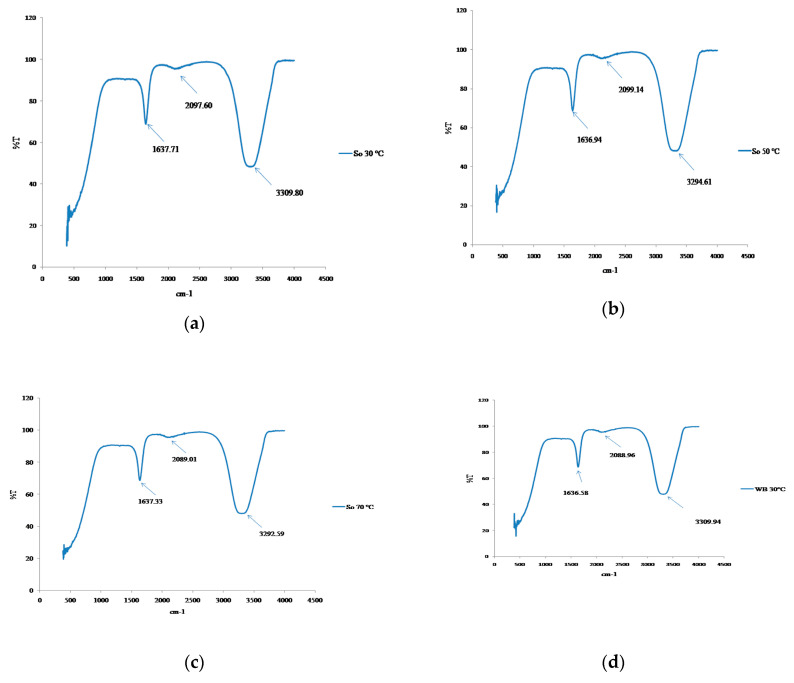
Fourier-transform infrared (FTIR) spectroscopy analysis of amaranth extract. (**a**) Ultra-sonication at 30 °C, (**b**) ultra-sonication at 50 °C, (**c**) ultra-sonication at 70 °C, (**d**) agitation in water bath (WB) at 30 °C, (**e**) agitation water bath (WB) at 50 °C, and (**f**) agitation water bath (WB) at 70 °C.

**Figure 6 foods-09-01116-f006:**
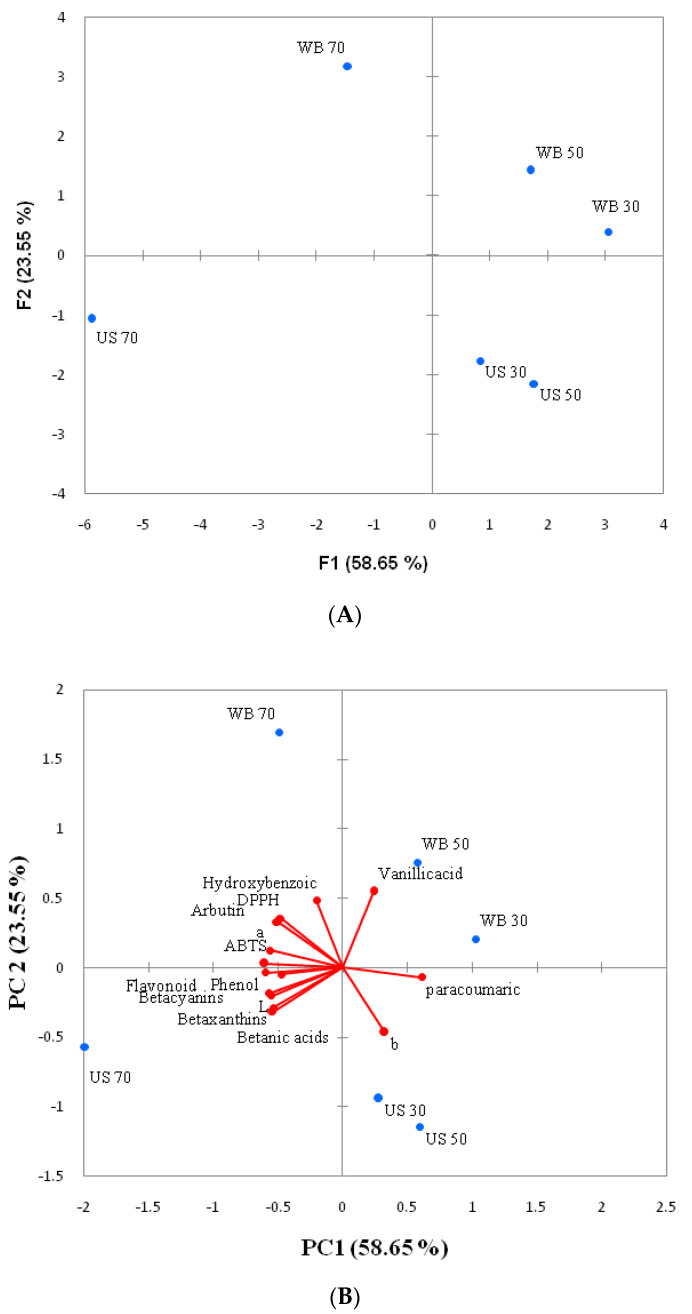
Principal component analysis of amaranth extract. (**A**) Parameter loading; (**B**) sample scores.

**Table 1 foods-09-01116-t001:** Effects of ultra-sonication and agitation on phenolic compounds(µg/g) of amaranth extract.

Parameters	Extraction Method					
	US 30	US 50	US 70	AG 30	AG 50	AG 70
Arbutin	^A^ 26485.60 ± 2544.45 ^a^	^A^ 26691.20 ± 1821.51 ^a^	^B^ 48458.40 ± 3920.77 ^c^	^A^ 19419.4 ± 1549.13 ^a^	^B^ 35148.4 ± 2588.01 ^b^	^C^ 53958.8 ± 5624.61 ^c^
Hydroxybenzoic acid	^AB^ 1817.20 ± 101.26 ^ab^	^A^ 1519.40 ± 135.48 ^a^	^B^ 1898.80 ± 110.31 ^ab^	^A^ 1654.2 ± 189.22 ^a^	^A^ 2183.88 ± 76.54 ^b^	^A^ 2153.2 ± 236.46 ^b^
Vanillic acid	^A^ 35.08 ± 4.92 ^a^	^A^ 31.72 ± 2.55 ^a^	^A^ 35.84 ± 1.33 ^a^	^A^ 158.4 ± 1.7 ^b^	^A^ 147.8 ± 91.36 ^b^	^A^ 146.6 ± 74.39 ^b^
*p*-coumaric acid	^A^ 57.84 ± 1.7 ^bc^	^A^ 57.48 ± 15.67 ^bc^	^A^ 28.80 ± 0.09 ^a^	^A^ 69.96 ± 7.86 ^c^	^A^ 61.4 ± 14.65 ^bc^	^A^ 39.36 ± 2.26 ^ab^
Ferulic acid	^A^ 79.84 ± 5.88 ^ab^	^A^ 77.68 ± 10.3 ^ab^	^A^ 76.00 ± 2.38 ^a^	^AB^ 85.16 ± 3.11 ^ab^	^A^ 73.8 ± 5.6 ^a^	^B^ 93.76 ± 7.01 ^b^
Total phenolic compounds	^A^ 28475.56 ± 2644.97 ^a^	^A^ 28377.45 ± 1980.36 ^a^	^C^ 50371.20 ± 3924.38 ^c^	^A^ 23657.24 ± 1481.36 ^a^	^B^ 36819.16 ± 1650.274 ^b^	^C^ 53311.12 ± 1569.55 ^c^

US30 = ultra-sonication at 30 °C; US50 = ultra-sonication at 50 °C; US70 = ultra-sonication at 70 °C; AG 30 = Agitation in water bath at 30 °C; AG 50 = Agitation in water bath at 50 °C; AG 70 = Agitation in water bath at 70 °C; ^a–c^ Means followed by different subscript alphabets are significantly different (*p* ≤ 0.05) between extraction method. ^A–C^ Means followed by different subscript alphabets are significantly different (*p* ≤ 0.05) among different temperatures in each extraction method.

**Table 2 foods-09-01116-t002:** Factor loading, eigenvalue, cumulative variance (%) and score for the first two principal (PC1. and PC2) components of amaranth extract based on extraction methods.

Loading	PC 1	PC2
L*	−0.871	−0.315
a*	−0.884	0.195
b*	0.516	−0.737
Phenol	−0.945	−0.056
Flavonoid	−0.747	−0.080
DPPH^+^	−0.756	0.562
ABTS^+^	−0.949	0.044
Betacyanins	−0.903	−0.289
Betaxanthins	−0.847	−0.453
Betanic acids	−0.851	−0.498
Arbutin	−0.795	0.521
Hydroxybenzoic	−0.317	0.762
Vanillicacid	0.379	0.863
Paracoumaric	0.966	−0.109
Ferulic acid	−0.005	0.616
**Scores**		
US 30	0.831	−1.771
US 50	1.767	−2.152
US 70	−5.891	−1.068
WB 30	3.042	0.390
WB50	1.704	1.420
WB70	−1.454	3.180

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
