# Peer review of "Effects of Ultra-Sonication and Agitation on Bioactive Compounds and Structure of Amaranth Extract"

_foods, 2020, doi:10.3390/foods9081116_

Round 1
Reviewer 1 Report
Generally, the paper have good experimental work but although the paper is well structured and the methodology correctly developed, the paper cannot be considered for publication in its actual format.
First of all, you need to pay attention to the formatting: there are words attached and there are several spaces missing. It is very difficult and tiring to read an article with all these writing errors. I have pointed out some of them in the comments, but a more careful revision is strongly suggested.
Abstract
Line 21: “In fact, higher temperature (70°C) showed greater amount of….”
Lines 22-23: “Meanwhile, temperature did not affect vanillic acid, p-coumaric acid and ferulic acid for both samples”
Introduction
In general, the introduction is too short. In particular, you have to add and described more papers about bioactive compounds from red amaranth. This suggestion is very important to make your paper more complete and interesting. Moreover, in my opinion the potential applications of red amaranth extracts should be emphasized.
Line 33: “… is a major source of vitamins…”
Line 34: “…amino acids [1].”
Line 36: “…diseases [2].”
Lines 40-41: “A few Amaranthus species have also shown strong Line 43antioxidant and anti-proliferative activity on Ehrlich’s ascites carcinoma cells [4].”
Line 43: “…various health…”
Line 45: “…as water, methanol and ethanol [5, 6, 7].”
Lines 59-60: add references
Materials and Methods
First of all, you have to correct all ABTS.+ and DPPH.+ in ABTS and DPPH or ABTS.+and DPPH.+. You decide
Line 71: “Fresh red amaranth (Amaranthus cruentus) was procured from the local market.”
You have to specify where the market is located. Moreover, how many plants do you utilize for this study?
Line 74: “The freshly cut small pieces (leafy parts) were…”
Lines 86-87: “In particular, 6.25 g of amaranth powder was mixed with 100 mL of distilled water in a 250 mL conical flask and placed in an ultra-sonicator bath at 30, 50, and 70°C for 5 min.
Lines 91-94: “Briefly, 6.25 g of amaranth powder was mixed with 100 mL of distilled water in a 250 mL conical flask and agitated using a shaking water bath (JSSB-50T, South Korea) at 100 rpm at 30, 50, and 70℃ for 5 min. After that, the samples were filtered through a cheese cloth, followed by vacuum filtration through Whatman filter paper No.1.”
Lines 96-97: I would change as following:
“Colour of the amaranth extracts was determined by using a Chroma meter (Minolta, CR-300, Osaka, Japan). Readings were expressed as L*, a* and b* parameters.”
Lines 99-101: “Amaranthus pigments were determined using the modified method described by Kumar et al. [6]. In particular, 1mL of aliquot was diluted with 9.0 mL of distilled water and absorbance was measured at 538, 480 and 430 nm for betacyanins, betaxanthins and betalamic acid, respectively, using a UV/Vis spectrophotometer (Optizen 2120 UV, Mecasys Co., South Korea).”
In Equation 1: ε and not € is the symbol for molar extinction coefficient. Moreover, you have to specify also L in lines 104-107.
Lines 104-107: “Where, A = Absorbance; MW = Molecular weight of betacyanins (726.6) betaxanthins (309) and betalamic acid (212); V = Solution volume; DF = Dilution factor; ε = Molar extinction coefficient of betacyanins (5.66 × 104 M-1 cm-1), betaxanthins (48000 M-1 cm-1), and betalamic acid (24000 M-1 cm-1); W = Sample weight (g).”
Line 115: “…as mg per gram sample (mg/g).”
Pay attention in the corresponding figure (Fig 2A) the results are expressed as µg/g. Which one is the right one?
Lines 117-118: “Briefly, 1 mL of 5-times diluted sample was treated with 4 mL distilled water and 0.3 mL of 5% sodium nitrite…”
Lines 122-123: “…as mg per gram sample (mg/g).”
Pay attention in the corresponding figure (Fig 2B) the results are expressed as µg/g. Which one is the right one?
Line 128: “The mobile phases was constituted by…”
Line 131: “(5 times dilution with water)”
Line 133: “…30°C. Chromatograms were recorded at 280 nm for arbutin, hydroxybenzoic and vanillic acid, and at 320 nm for p-coumaric and ferulic acid.
Lines 137-138: “…as mg per gram sample (mg/g).”
Pay attention in the corresponding table (Table 1) and in text (Section 3.4 in Results and Discussion) the results are expressed as µg/g. Which one is the right one?
Line 141-146: “In particular, 24 mg of DPPH was prepared in methanol and used as a stock solution. The working solution was prepared by diluting 10 mL of the stock solution with methanol. Amaranth extract (1 mL) was diluted 5 times with water and 200 μL samples were mixed with 2 mL of the DPPH solution. After 30 min, the absorbance was measured at 515 nm using UV/Vis spectrophotometer. The DPPH radical-scavenging activity was calculated as follows:”
Line 151: “Briefly, 7.4 mM ABTS…”
Line 157: “…and A1 = absorbance…”
Line 167: Were the results expressed as mean vale ± standard deviation (SD)? Please add here.
Lines 169-170: “(SPSS for Windows Version 21.0). Principal Component Analysis (PCA)…”
Results and discussion
In general, all figures must be improved. First, the background of all figures must be white and not coloured: in this way the figure looks better and more fair. Then pay attention because some figures are crushed, others enlarged too much. Please make them all uniform. Moreover, check carefully spaces missing in the text captions of all figures.
Lines 173-174: “L*, a* and b* values of amaranth extracts obtained by means of ultra-sonication and agitation at different temperatures were shown in Figure 1. Higher L* values were observed with ultra-sonication samples than…”
Line 177: “…did not have…”
Line 180: “…forms such as betaxanthin and betalamic acid. A higher…”
Lines 183-185: “However, no correlation was observed between a* and b* values and the amaranth pigments. While L* and a* values were much lower, b* values were found to be much more higher than that of powders derived from different parts of the amaranthus species [3].”
Lines 185-186: you have written that thee deviation may be due to variation in cultivar, and processing environments. Please add references that stated these differences.
Line 197: 3.2. Effects of ultra-sonication and agitation on antioxidant properties
In this paragraph you have to correct all ABTS.+ and DPPH.+ in ABTS and DPPH or ABTS.+and DPPH.+, as previously suggested
Line 199: “…were shown in Figures 2 and 3.
Line 201: “…did not significantly affect…”
Line 204: “were obtained” instead of “found”
Line 206: “…(70℃) might be related…”
Lines 207-208: add references
Lines 208-209: “…temperature. The increase in polyphenolic oxidase activity…”
Line 211: “various juices [18,19] and…”
Line 219: “(r value of 0.78 at p<0.05). These results…”
Line 222: “…acting as stronger…”
Moreover, add references.
Line 242: “…respectively for…”
Line 244: “…from 4536.59 to…”
Line 246: add references
Line 247: “…amaranthus seed…”
Line 249: “…to 6.02 mg/100g) and amaranthus sprouts (2.69 mg/100 g) [3]. These…”
Line 250: add references
Line 253: “…employed. This…”
Line 255: “…did not have significant effect on amaranth pigments. The three pigments followed…”
Lines 262-263: correct ABTS as previously suggested
Line 275: in italics
In this paragraph please add at least the best chromatograms of the two extraction methods considered in this paper (ultra-sonication and agitation)
Lines 276-277: “…in amaranthus extracts as influenced by ultra-sonication and agitation were shown in Table 1.”
Lines 278-280: “…were arbutin > hydroxybenzoic acid > ferulic acid > p- coumaric acid > vanillic acid whereas agitated samples contained arbutin > hydroxybenzoic acid > vanillic acid > ferulic acid > p- coumaric acid.”
Line 282: “…and quercetin…”
Line 284: “…syringic phenolic acids were…”
Line 285: “…methods and…”
Lines 287-288: “…(70°C) significantly increased arbutin and hydroxybenzoic acid as compared to lower temperature (30°C) in ultra-sonication methods…”
Line 290: “…acid and ferulic…”
Line 291: “…methods. As…”
Lines 292-293: add references
Line 294: “…agitation [18, 19].”
Line 295: “…[23]. In a…”
Line 297: “…to 30°C. In this…”
Line 299: “…acid (28.80 to 69.91 μg/g) and hydroxybenzoic acid (1519.40 to 2183.88 μg/g) were…”
Line 303: “…amaranth. Nonetheless…”
In Table 1 check carefully spaces missing in the footnotes
Line 310: 3.5. Structural changes
Line 311: “…by ultra-sonication…”
Line 312: “…were shown in Figure 5 a-f…”
Line 313: “…are considered…”
Lines 314-315: “…at 1636.58 to1637 cm−1 and 2089 to 2125 cm−1 for all amaranth extract samples. These bands indicate the functional group of betacyanin.”
Line 316: “…that 1653 to 918 cm−1…”
Line 316: bands represent or bands represent?
Line 318: “These variations…”
Line 319: bands…were found or band…was found?
Lines 331-333: please correct the text captions of Figure 5: there are many formatting and writing errors
Line 334: 3.6. Principal component analysis (PCA)
In this paragraph, you have to correct always in PC1 e in PC2
Line 338: “…phenols, flavonoids…”
Line 339 and 342: correct ABTS and DPPH as previously suggested
Line 341: “…to the amount…”
Lines 341-343: compounds should not be written in capital letters
Line 343: “On the other hand higher positive scores were associated…”
Line 346: “…that could be…”
Conclusion
Line 358: “…investigated. No…”
Line 359: “…total phenols, total flavonoids…”
Moreover, correct ABTS and DPPH as previously suggested
Lines 360-362: “…acid of amaranth extracts prepared using ultra-sonication and agitation. Higher temperature led to increased total phenolic compounds in the amaranth extracts…”
Line 363: “…(PCA) was…”
Line 364: “…agitation extraction at 70°C could be…”
Please you should format the references according to the instructions for the authors suggested by MDPI
Author Response
Open Review
Abstract
Line 21: “In fact, higher temperature (70°C) showed greater amount of….”
Corrected as reviewer comments and mentioned it in text.
Lines 22-23: “Meanwhile, temperature did not affect vanillic acid, p-coumaric acid and ferulic acid for both samples”
Corrected as reviewer comments and mentioned it in text.
Introduction
In general, the introduction is too short. In particular, you have to add and described more papers about bioactive compounds from red amaranth. This suggestion is very important to make your paper more complete and interesting. Moreover, in my opinion the potential applications of red amaranth extracts should be emphasized.
Corrected as reviewer comments and mentioned it in text.
Line 33: “… is a major source of vitamins…”
Corrected as reviewer comments and mentioned it in text.
Line 34: “…amino acids [1].”
Corrected as reviewer comments and mentioned it in text.
Line 36: “…diseases [2].”
Corrected as reviewer comments and mentioned it in text.
Lines 40-41: “A few Amaranthus species have also shown strong Line 43antioxidant and anti-proliferative activity on Ehrlich’s ascites carcinoma cells [4].”
Corrected as reviewer comments and mentioned it in text.
Line 43: “…various health…”
Corrected as reviewer comments and mentioned it in text.
Line 45: “…as water, methanol and ethanol [5, 6, 7].”
Corrected as reviewer comments and mentioned it in text.
Lines 59-60: add references
Corrected as reviewer comments and mentioned it in text.
Materials and Methods
First of all, you have to correct all ABTS.+ and DPPH.+ in ABTS and DPPH or ABTS.+and DPPH.+. You decide
Corrected as reviewer comments and mentioned it in through out text.
Line 71: “Fresh red amaranth (Amaranthus cruentus) was procured from the local market.”
Corrected as reviewer comments and mentioned it in text.
You have to specify where the market is located. Moreover, how many plants do you utilize for this study?
Corrected as reviewer comments and mentioned it in text.
Line 74: “The freshly cut small pieces (leafy parts) were…”
Corrected as reviewer comments and mentioned it in text.
Lines 86-87: “In particular, 6.25 g of amaranth powder was mixed with 100 mL of distilled water in a 250 mL conical flask and placed in an ultra-sonicator bath at 30, 50, and 70°C for 5 min.
Corrected as reviewer comments and mentioned it in text.
Lines 91-94: “Briefly, 6.25 g of amaranth powder was mixed with 100 mL of distilled water in a 250 mL conical flask and agitated using a shaking water bath (JSSB-50T, South Korea) at 100 rpm at 30, 50, and 70℃ for 5 min. After that, the samples were filtered through a cheese cloth, followed by vacuum filtration through Whatman filter paper No.1.”
Corrected as reviewer comments and mentioned it in text.
Lines 96-97: I would change as following:
Corrected as reviewer comments and mentioned it in text.
“Colour of the amaranth extracts was determined by using a Chroma meter (Minolta, CR-300, Osaka, Japan). Readings were expressed as L*, a* and b* parameters.”
Corrected as reviewer comments and mentioned it in text.
Lines 99-101: “Amaranthus pigments were determined using the modified method described by Kumar et al. [6]. In particular, 1mL of aliquot was diluted with 9.0 mL of distilled water and absorbance was measured at 538, 480 and 430 nm for betacyanins, betaxanthins and betalamic acid, respectively, using a UV/Vis spectrophotometer (Optizen 2120 UV, Mecasys Co., South Korea).”
Corrected as reviewer comments and mentioned it in text.
In Equation 1: ε and not € is the symbol for molar extinction coefficient. Moreover, you have to specify also L in lines 104-107.
Corrected as reviewer comments and mentioned it in text.
Lines 104-107: “Where, A = Absorbance; MW = Molecular weight of betacyanins (726.6) betaxanthins (309) and betalamic acid (212); V = Solution volume; DF = Dilution factor; ε = Molar extinction coefficient of betacyanins (5.66 × 104 M-1 cm-1), betaxanthins (48000 M-1 cm-1), and betalamic acid (24000 M-1 cm-1); W = Sample weight (g).”
Corrected as reviewer comments and mentioned it in text.
Line 115: “…as mg per gram sample (mg/g).”
Corrected as reviewer comments and mentioned it in text.
Pay attention in the corresponding figure (Fig 2A) the results are expressed as µg/g. Which one is the right one?
Corrected as reviewer comments and mentioned it in text.
Lines 117-118: “Briefly, 1 mL of 5-times diluted sample was treated with 4 mL distilled water and 0.3 mL of 5% sodium nitrite…”
Corrected as reviewer comments and mentioned it in text.
Lines 122-123: “…as mg per gram sample (mg/g).”
Corrected as reviewer comments and mentioned it in text.
Pay attention in the corresponding figure (Fig 2B) the results are expressed as µg/g. Which one is the right one?
Corrected as reviewer comments and mentioned it in text.
Line 128: “The mobile phases was constituted by…”
Corrected as reviewer comments and mentioned it in text.
Line 131: “(5 times dilution with water)”
Corrected as reviewer comments and mentioned it in text.
Line 133: “…30°C. Chromatograms were recorded at 280 nm for arbutin, hydroxybenzoic and vanillic acid, and at 320 nm for p-coumaric and ferulic acid.
Corrected as reviewer comments and mentioned it in text.
Lines 137-138: “…as mg per gram sample (mg/g).”
Corrected as reviewer comments and mentioned it in text.
Pay attention in the corresponding table (Table 1) and in text (Section 3.4 in Results and Discussion) the results are expressed as µg/g. Which one is the right one?
Corrected as reviewer comments and mentioned it in text.

Reviewer 2 Report
Comments to the Authors:
The authors of this paper present a study on the effects of ultra-sonication and agitation on bioactive compounds and structure of amaranth extract.
Since this study may give an additional economical and nutraceutical value to a plant extract the topic may be interesting.
During the initial checking of the manuscript, we saw that there are few overlaps in your manuscript (mainly from author’s previews papers). That means the highlighted parts( see attached turnitin file) is similar to some published works. Please kindly revise/refresh these parts in your manuscript.
- Abstract:
-Please explain that you refer to Hunter color values (L*, a*, b*) when in first time you report these values.
-Please follow the journal format (Background –methods - results – conclusion) and add some numerical results.
-Please report the DPPH.+ radical as DPPH· and ABTS.+ as ABTS.+ in all manuscript.
- Introduction:
-Please correct all manuscript since many words are without the appropriate gap (e.g. Line 30 majorsource)
-Line 59-60 : Avoid making factual statements without any references .For example “However, studies have shown that many synthetic food colorants likely to be carcinogenic to consumers” Add a reference.
-Line 50:The introduction in on UAE could perhaps be strengthened by describing also some benefits of this non-conventional extraction method) See Skenderidis et al. Optimization of Ultrasound Assisted Extraction ……
- Materials and Methods:
-Please add the methodology for Hunter (L∗, a∗, and b∗) color values measurement.
- Results/Discussion:
-The presentation of FTIR is limited are not discussed accordingly .The authors present only in separate not legible figures the % transmittance of each sample and there is no analysis of the presented peaks and no comparisons performed in the fingerprint region. You can check relative paper with FTIR analysis ultra sound assisted extracts “The in vitro antimicrobial activity assessment of ultrasound assisted…..”

Author Response
- Abstract:
-Please explain that you refer to Hunter color values (L*, a*, b*) when in first time you report these values.
Corrected as reviewer comments and mentioned it in text.
-Please follow the journal format (Background –methods - results – conclusion) and add some numerical results.
Corrected as reviewer comments and mentioned it in text.
-Please report the DPPH.+ radical as DPPH· and ABTS.+ as ABTS.+ in all manuscript.
Corrected as reviewer comments and mentioned it in text.
Introduction:
-Please correct all manuscript since many words are without the appropriate gap (e.g. Line 30 majorsource)
Corrected as reviewer comments and mentioned it in text.
-Line 59-60 : Avoid making factual statements without any references .For example “However, studies have shown that many synthetic food colorants likely to be carcinogenic to consumers” Add a reference.
Corrected as reviewer comments and mentioned it in text.
-Line 50:The introduction in on UAE could perhaps be strengthened by describing also some benefits of this non-conventional extraction method) See Skenderidis et al. Optimization of Ultrasound Assisted Extraction ……
Corrected as reviewer comments and mentioned it in text.
- Materials and Methods:
-Please add the methodology for Hunter (L∗, a∗, and b∗) color values measurement.
Corrected as reviewer comments and mentioned it in text.
- Results/Discussion:
-The presentation of FTIR is limited are not discussed accordingly .The authors present only in separate not legible figures the % transmittance of each sample and there is no analysis of the presented peaks and no comparisons performed in the fingerprint region. You can check relative paper with FTIR analysis ultra sound assisted extracts “The in vitro antimicrobial activity assessment of ultrasound assisted…..”
Corrected as reviewer comments and mentioned it in text. However we did not add the finger print region. We believe and saw many researchers mentioned that % transmittance might be sufficient to describe the FTIR methods.
Line 141-146: “In particular, 24 mg of DPPH was prepared in methanol and used as a stock solution. The working solution was prepared by diluting 10 mL of the stock solution with methanol. Amaranth extract (1 mL) was diluted 5 times with water and 200 μL samples were mixed with 2 mL of the DPPH solution. After 30 min, the absorbance was measured at 515 nm using UV/Vis spectrophotometer. The DPPH radical-scavenging activity was calculated as follows:”
Corrected as reviewer comments and mentioned it in text. Line 151: “Briefly, 7.4 mM ABTS…”
Line 157: “…and A1 = absorbance…”
Corrected as reviewer comments and mentioned it in text.
Line 167: Were the results expressed as mean vale ± standard deviation (SD)? Please add here.
Corrected as reviewer comments and mentioned it in text.
Lines 169-170: “(SPSS for Windows Version 21.0). Principal Component Analysis (PCA)…”
Corrected as reviewer comments and mentioned it in text.
Results and discussion
In general, all figures must be improved. First, the background of all figures must be white and not coloured: in this way the figure looks better and more fair. Then pay attention because some figures are crushed, others enlarged too much. Please make them all uniform. Moreover, check carefully spaces missing in the text captions of all figures.
Corrected as reviewer comments and mentioned it in text.
Lines 173-174: “L*, a* and b* values of amaranth extracts obtained by means of ultra-sonication and agitation at different temperatures were shown in Figure 1. Higher L* values were observed with ultra-sonication samples than…”
Corrected as reviewer comments and mentioned it in text.
Line 177: “…did not have…”
Corrected as reviewer comments and mentioned it in text.
Line 180: “…forms such as betaxanthin and betalamic acid. A higher…”
Corrected as reviewer comments and mentioned it in text.
Lines 183-185: “However, no correlation was observed between a* and b* values and the amaranth pigments. While L* and a* values were much lower, b* values were found to be much more higher than that of powders derived from different parts of the amaranthus species [3].”
Corrected as reviewer comments and mentioned it in text.
Lines 185-186: you have written that thee deviation may be due to variation in cultivar, and processing environments. Please add references that stated these differences.
Corrected as reviewer comments and mentioned it in text.
Line 197: 3.2. Effects of ultra-sonication and agitation on antioxidant properties
Corrected as reviewer comments and mentioned it in text.
In this paragraph you have to correct all ABTS.+ and DPPH.+ in ABTS and DPPH or ABTS.+and DPPH.+, as previously suggested
Corrected as reviewer comments and mentioned it in text.
Line 199: “…were shown in Figures 2 and 3.
Corrected as reviewer comments and mentioned it in text.
Line 201: “…did not significantly affect…”
Corrected as reviewer comments and mentioned it in text.
Line 204: “were obtained” instead of “found”
Corrected as reviewer comments and mentioned it in text.
Line 206: “…(70℃) might be related…”
Corrected as reviewer comments and mentioned it in text.
Lines 207-208: add references
Corrected as reviewer comments and mentioned it in text.
Lines 208-209: “…temperature. The increase in polyphenolic oxidase activity…”
Corrected as reviewer comments and mentioned it in text.
Line 211: “various juices [18,19] and…”
Corrected as reviewer comments and mentioned it in text.
Line 219: “(r value of 0.78 at p<0.05). These results…”
Corrected as reviewer comments and mentioned it in text.
Line 222: “…acting as stronger…”
Corrected as reviewer comments and mentioned it in text.
Moreover, add references.
Corrected as reviewer comments and mentioned it in text.
Line 242: “…respectively for…”
Corrected as reviewer comments and mentioned it in text.
Line 244: “…from 4536.59 to…”
Corrected as reviewer comments and mentioned it in text.
Line 246: add references
Corrected as reviewer comments and mentioned it in text.
Line 247: “…amaranthus seed…”
Corrected as reviewer comments and mentioned it in text.
Line 249: “…to 6.02 mg/100g) and amaranthus sprouts (2.69 mg/100 g) [3]. These…”
Corrected as reviewer comments and mentioned it in text.
Line 250: add references
Corrected as reviewer comments and mentioned it in text.
Line 253: “…employed. This…”
Corrected as reviewer comments and mentioned it in text.
Line 255: “…did not have significant effect on amaranth pigments. The three pigments followed…”
Corrected as reviewer comments and mentioned it in text.
Lines 262-263: correct ABTS as previously suggested
Corrected as reviewer comments and mentioned it in text.
Line 275: in italics
Corrected as reviewer comments and mentioned it in text.
In this paragraph please add at least the best chromatograms of the two extraction methods considered in this paper (ultra-sonication and agitation)
Corrected as reviewer comments and mentioned it in text.
Lines 276-277: “…in amaranthus extracts as influenced by ultra-sonication and agitation were shown in Table 1.”
Corrected as reviewer comments and mentioned it in text.
Lines 278-280: “…were arbutin > hydroxybenzoic acid > ferulic acid > p- coumaric acid > vanillic acid whereas agitated samples contained arbutin > hydroxybenzoic acid > vanillic acid > ferulic acid > p- coumaric acid.”
Corrected as reviewer comments and mentioned it in text.
Line 282: “…and quercetin…”
Corrected as reviewer comments and mentioned it in text.
Line 284: “…syringic phenolic acids were…”
Corrected as reviewer comments and mentioned it in text.
Line 285: “…methods and…”
Corrected as reviewer comments and mentioned it in text.
Lines 287-288: “…(70°C) significantly increased arbutin and hydroxybenzoic acid as compared to lower temperature (30°C) in ultra-sonication methods…”
Corrected as reviewer comments and mentioned it in text.
Line 290: “…acid and ferulic…”
Corrected as reviewer comments and mentioned it in text.
Line 291: “…methods. As…”
Corrected as reviewer comments and mentioned it in text.
Lines 292-293: add references
Corrected as reviewer comments and mentioned it in text.
Line 294: “…agitation [18, 19].”
Corrected as reviewer comments and mentioned it in text.
Line 295: “…[23]. In a…”
Corrected as reviewer comments and mentioned it in text.
Line 297: “…to 30°C. In this…”
Corrected as reviewer comments and mentioned it in text.
Line 299: “…acid (28.80 to 69.91 μg/g) and hydroxybenzoic acid (1519.40 to 2183.88 μg/g) were…”
Corrected as reviewer comments and mentioned it in text.
Line 303: “…amaranth. Nonetheless…”
Corrected as reviewer comments and mentioned it in text.
In Table 1 check carefully spaces missing in the footnotes
Corrected as reviewer comments and mentioned it in text.
Line 310: 3.5. Structural changes
Corrected as reviewer comments and mentioned it in text.
Line 311: “…by ultra-sonication…”
Corrected as reviewer comments and mentioned it in text.
Line 312: “…were shown in Figure 5 a-f…”
Corrected as reviewer comments and mentioned it in text.
Line 313: “…are considered…”
Corrected as reviewer comments and mentioned it in text.
Lines 314-315: “…at 1636.58 to1637 cm−1 and 2089 to 2125 cm−1 for all amaranth extract samples. These bands indicate the functional group of betacyanin.”
Corrected as reviewer comments and mentioned it in text.
Line 316: “…that 1653 to 918 cm−1…”
Corrected as reviewer comments and mentioned it in text.
Line 316: bands represent or bands represent?
Corrected as reviewer comments and mentioned it in text.
Line 318: “These variations…”
Corrected as reviewer comments and mentioned it in text.
Line 319: bands…were found or band…was found?
Corrected as reviewer comments and mentioned it in text.
Lines 331-333: please correct the text captions of Figure 5: there are many formatting and writing errors
Corrected as reviewer comments and mentioned it in text.
Line 334: 3.6. Principal component analysis (PCA)
Corrected as reviewer comments and mentioned it in text.
In this paragraph, you have to correct always in PC1 e in PC2
Corrected as reviewer comments and mentioned it in text.
Line 338: “…phenols, flavonoids…”
Corrected as reviewer comments and mentioned it in text.
Line 339 and 342: correct ABTS and DPPH as previously suggested
Corrected as reviewer comments and mentioned it in text.
Line 341: “…to the amount…”
Corrected as reviewer comments and mentioned it in text.
Lines 341-343: compounds should not be written in capital letters
Corrected as reviewer comments and mentioned it in text.
Line 343: “On the other hand higher positive scores were associated…”
Corrected as reviewer comments and mentioned it in text.
Line 346: “…that could be…”
Corrected as reviewer comments and mentioned it in text.
Conclusion
Line 358: “…investigated. No…”
Corrected as reviewer comments and mentioned it in text.
Line 359: “…total phenols, total flavonoids…”
Corrected as reviewer comments and mentioned it in text.
Corrected as reviewer comments and mentioned it in text.
Moreover, correct ABTS and DPPH as previously suggested
Corrected as reviewer comments and mentioned it in text.
Lines 360-362: “…acid of amaranth extracts prepared using ultra-sonication and agitation. Higher temperature led to increased total phenolic compounds in the amaranth extracts…”
Corrected as reviewer comments and mentioned it in text.
Line 363: “…(PCA) was…”
Corrected as reviewer comments and mentioned it in text.
Line 364: “…agitation extraction at 70°C could be…”
Corrected as reviewer comments and mentioned it in text.
Please you should format the references according to the instructions for the authors suggested by MDPI
Corrected as reviewer comments and mentioned it in text.
